# The *TTN* p. Tyr4418Ter mutation causes cardiomyopathy in human and mice

**Wenqiang Sun**[1☯], **Xiaohui Liu**[2☯], **Laichun Song**[1], **Liang Tao**[1], **Kaisheng Lai**[3], **Hui Jiang**[3], **Hongyan Xiao**[1]*

1 Division of Cardiac Surgery & Wuhan Clinical Research Center for Cardiomyopathy, Wuhan Asia Heart Hospital Affiliated with Wuhan University of Science and Technology, Wuhan, Hubei, P.R. China, 2 Department of Clinical Laboratory, Wuhan Asia Heart Hospital Affiliated with the Wuhan University of Science and Technology, Wuhan, Hubei, P.R. China, 3 Department of Science Research Centre, BestNovo (Beijing) Medical Laboratory, Beijing, P.R. China

☯ These authors contributed equally to this work.

* xiaohy11@sina.com

**Data Availability Statement:** The sequencing data has been submitted to NCBI SRA database with the accession number PRJNA977831 and can be

## Abstract

### Objective

To generate a mouse model carrying *TTN*tv Y4370* simulating the newly discovered human heterozygous nonsense *TTN*tv c.13254T>G (p.Tyr4418Ter) to supplement and improve the functional evidence of pathogenic mutation *TTN*tv c.13254T>G on the pathogenic type of dilated cardiomyopathy.

### Methods

We generated 4 mice carrying *TTN*tv p. Y4370* through CRISPR/Cas-mediated genome engineering. Monthly serological detection, bimonthly echocardiography, and histology evaluation were carried out to observe and compare alterations of cardiac structure and function between 4 *TTN*[+/-] mice and 4 wild-type (WT) mice.

### Results

For the two-month-old *TTN*[+/-] mice, serum glutamic-oxalacetic transaminase (AST), lactic dehydrogenase (LDH), and creatine kinase (CK) were significantly increased, the diastolic Left Ventricular Systolic Anterior Wall (LVAW), and the LV mass markedly rose, with the left ventricular volume displaying an increasing trend and Ejection Fraction (EF) and Fractional Shortening (FS) showing a decreasing trend. Besides, the histological evaluation showed that cardiac fibrosis level and positive rate of cardiac mast cell of *TTN*[+/-] mice were obviously increased compared with WT mice.

### Conclusions

*TTN*tv Y4370* could lead to cardiac structure and function alterations in mice, supplementing the evidence of *TTN*tv c.13254T>G pathogenicity in human.

accessed with the URL https://www.ncbi.nlm.nih.gov/sra/PRJNA977831.

**Funding:** This study is supported by Hubei Provincial Natural Science Foundation of China (2022CFB841) and Scientific Research Program of Wuhan Municipal Health Commission (WX20D15). There was no other additional external funding received for this study. The author Wenqiang Sun received both grants. Dr Wenqiang Sun investigated and analyzed the data of this study, and wrote the original manuscript and reviewed the logic of the manuscript. In addition, Dr Wenqiang Sun supervised and reviewed the whole research process.

**Competing interests:** The authors declare no competing interests.

## Introduction

The titin encoded by the *TTN* gene with 363 coding exons is the largest protein in the human body and is an important determinant of the contractile, elastic, and signal properties of the heart and skeletal muscle [1, 2]. Therefore, *TTN* variants are associated with different genetic diseases, including hypertrophic and dilated cardiomyopathy and several skeletal muscle diseases [3, 4]. Among them, dilated cardiomyopathy (DCM) characterized by left ventricular dilation, increased left ventricular volume and decreased ejection fraction is the main indication for heart transplantation, affecting about 1:250 individuals [5, 6]. There is a wide range of genetic heterogeneity in DCM [7], and at least 39 gene variants contribute to this phenotype [8], while *TTN* truncating variant (*TTN*tv) is the most common genetic cause of DCM, accounting for about 25% of familial DCM cases and 18% of sporadic DCM cases [9]. *TTN*tv in DCM patients is mainly located in the sarcomere A-band of the protein, which occurs near the s-carboxyl end of the protein, and exists in a large number of transcribed exons in the myocardium [10, 11].

Compared with other known DCM pathogenic genes, research on the effects of *TTN* mutations is relatively late. There are two reasons why fewer *TTN* mutations have been discovered. Firstly, the large *TTN* molecules make it difficult for traditional molecular biology methods to detect mutations. Secondly, there are numerous subtypes of *TTN* proteins that undergo selective cleavage to produce various skeletal and myocardial subtypes, including N2B, N2BA, and NOVEX-3 [9, 12] and are regulated by RNA binding motif protein (RBM) genes, which are associated with approximately 2% of DCM [13, 14]. Further research has shown that there are multiple sub-shear pathways capable of producing N2BA subtypes, and the heterogeneity of shear pathways is more complex in the PEVK region of the sarcomere [14]. In recent years, next-generation sequencing (NGS) has discovered many new *TTN* mutations and gradually recognized *TTN* as an important gene in human genetic diseases [15]. Due to its large size, *TTN*tv is also relatively common among the general population. A recent study by Pugh et al. [16] confirmed the presence of *TTN*tv (1.65%) in the general population and demonstrated that variants with truncated A-bands are more common in DCM patients than in the control group, indicating that explaining the pathological mechanism of *TTN* mutations is difficult.

A novel nonsense *TTN*tv c.13254T>G (*TTN* p. Tyr4418Ter) which hasn't been recorded in general populations according to the Varsome database was found in two patients with DCM. The *TTN* gene of mice (GenBank accession number: XM_036160543.1; Ensembl: ENSMUSG00000051747) is located in chromosome 2 with 45 identified exons, of which mutation Y4370* on exon 45 is consistent with the newly explored human *TTN* mutation Y4418*. Therefore, we constructed a C57BL/6J mouse model carrying Y4370* mutation by CRISPR/Cas-mediated genome engineering to imitate the newly discovered human *TTN*tv followed by clinical examinations, including echocardiography, serological detection and histological evaluation, which supplemented and improved the functional evidence of pathogenic mutation *TTN* c.13254T>G on the pathogenic type of dilated cardiomyopathy.

## Materials and methods

### Ethical statement

The study was conducted in accordance with The Code of Ethics of the World Medical Association. The Human Research Ethics Committee of the Wuhan Asia Heart Hospital approved the protocol for WES (ethics approval number: 2023-B019) and all subjects provided written informed consent. The mouse studies were performed in accordance with ethical guidelines of Wuhan Asia Heart Hospital, China. The experimental protocols were in accordance with the

ARRIVE guidelines and were approved by the Wuhan Asia Heart Hospital Animal Care and Use Committee (Ethics reference number: 2021-YXKY-P007). All animal protocols conformed to the National Research Council's Guide for the Care and Use of Laboratory Animals.

## Whole exome sequencing

Whole exome sequencing was performed for genetic analysis. KAPA DNA Library Preparation Kits for Illumina (US, Kapa Biosystems, Inc.) were used to construct the library, and IDT's liquid chip capture system was used to enrich the whole exon region of genomic DNA extracted from peripheral blood. Then high-throughput and high-depth sequencing was performed on the Illumina Novaseq platform.

## Generation of mouse model with *TTN* p.Y4370*

Sequence
(`GATGATGTTGGATATCATGGACCGGACTGGGGAAATATGAAAGGACATTCTCAAAGTGAT` (`TAT`) `GTGCTAAAT` (`ATC`) `CACTC//`
`CAAGAGAACCTCTAACACAGTTCAGGACGTGGAAGACTCACCTGTCCCTACCTAT`, "//" indicates gRNA cut site) on exon 45 of the mouse *TTN* gene was selected as the target site. gRNA-A1 sequence `TGTTAGAGGTTCTCTTGGAG−TGG`, and donor oligo
`GATGATGTTGGATATCATGGACCGGACTGGGGAAATATGAAAGGACATTCTCAAAGTGAT` (`TAG`) `GTGCTAAAT` (`ATA`)
`CACTCCAAGAGAACCTCTAACACAGTTCAGGACGTGGAAGACTCACCTGTCCCTACCTAT`
with targeting sequence flanked by 120 bp homologous sequences combined on both sides were designed. Off-target analysis for gRNA-A1 was performed. The p.Y4370* (TAT to TAG) in donor oligo was introduced into exon 45 by homology-directed repair. Synonymous mutation p. I4374I = (ATC> ATA) was introduced simultaneously to prevent gRNA binding and re-cutting the sequence after homology-directed repair. Cas9, gRNA and donor oligo were co-injected into fertilized eggs of mice with C57Bl/6J background for F0 mouse production. The pups were genotyped by PCR with specific forward primer `TTTCAAAGCCTTGAGACAGTTG` and reverse primer `GTTAGAGGAATCTGCTTCTGTGT`. A positive mouse with *TTN* p.Y4370* heterozygous mutation was selected to backcross with the C57Bl/6J background mouse for the production of F1 generation followed by genotyping. 4 *TTN*[+/-] mice (1 male and 3 females) of F1 generation and 4 even-aged C57Bl/6J wild-type (WT) mice (1 male and 3 females) were fed in the specific pathogen-free (SPF) environment for 10 months and subjected to serological detection, echocardiography and histological evaluation at different time points.

## Serological detection for mice

200 μL blood was taken from the eye vein of the *TTN*[+/-] mice and WT mice once a month for serological detection. The serum aspartate transaminase (AST), creatine kinase (CK), and lactate dehydrogenase (LDH) were examined by ELISA kit (Abcam, UK) for the first 7 times. In addition to AST, CK, and LDH, cTnI and CK-MB were also examined in the last serological detection. The first blood collection of the mice was less to ensure that the mice were healthy and normal.

## Echocardiography for mice

Echocardiography was performed by VINNO portable color ultrasound instrument every two months. The *TTN*[+/-] mice and WT mice were inhaled with a mixture of 1.5% ~ 2% isoflurane and oxygen (visual sonics, Canada) in the anesthesia room and fixed on a constant

temperature platform in the supine position to keep body temperature at 37±1 ˚C. After removing the chest hair, the ultrasonic probe was placed in the left anterior chest to obtain two-dimensional M-mode echocardiography. At least 6 consecutive cardiac cycles were monitored and recorded. Ejection fraction (EF) and fractional shortening (FS) were obtained to evaluate left ventricular function. PV velocity time integral (VTI), mean velocity (mean vel), mean grad, peak velocity (peak Vel), peak grad, and PAT were obtained to describe pulmonary flow.

## Masson staining

Under anesthesia, mice were humanely euthanized by cervical vertebrae dislocation. Hearts were harvested, rinsed with pre-cooled phosphate-buffered saline (PBS) to remove blood, and weighted. A portion of the heart tissue was fixed in 4% paraformaldehyde at room temperature for 48 hours. Subsequently, the heart tissue was embedded in paraffin and sliced. The samples were immersed in xylene I for 5 min, xylene II for 5 min, xylene III for 5 min, anhydrous ethanol for 1 min, 95% ethanol for 1 min, 75% ethanol for 1 min, and rinsed with tap water for several seconds. The samples were then stained with the prepared Weigert iron hematoxylin staining solution for 8 minutes. After that, the samples were differentiated with acidic ethanol differentiation solution for 15 seconds and washed with water. Subsequently, the samples were blue-backed for 5 minutes with Masson blue solution, and washed with distilled water for 1 minute after washing. The samples were stained with ponceau magenta staining solution for 5 minutes. Afterwards, the samples were successively washed with weak acid working solution, phosphomolybdic acid solution, and weak acid working solution for 1 minute each. The samples were stained with aniline blue staining solution for 2 minutes following by washing with weak acid for 1 minute. The samples were placed in 95% ethanol for rapid dehydration of 2–3 seconds, dehydrated with anhydrous ethanol for 3 times, 5–10 seconds each time, and then transparent with xylene for 3 times, 1–2 minutes each time, and finally sealed with neutral gum.

## Toluidine blue staining

The mouse heart tissue slice samples were placed in an oven and baked at 65 degrees Celsius for 30 minutes. The dried samples were placed on a staining rack and immersed in xylene I and xylene II solutions for five minutes each for dewaxing. The dewaxed samples were first immersed in anhydrous ethanol I for two minutes, and then immersed in 95% ethanol, 85% ethanol, 75% ethanol, and tap water for two minutes each. The hydrated sample slices were immersed with toluidine blue staining solution for 5–30 minutes. After staining, samples were rinsed thoroughly with running water for two minutes. Then 1% hydrochloric acid alcohol dropwise was added for differentiation until the cells were visible in a purple-blue color. After the differentiation was completed, the samples were rinsed with running water for 5 minutes, and the samples were rinsed with distilled water for 1–2 seconds. Then the rinsed slices were placed in an oven and thoroughly dried. The samples were immersed in anhydrous ethanol I for 30 seconds and anhydrous ethanol II for 1 minute to dehydrate the samples after dyeing and drying. The dehydrated toluidine blue stained slices were Immersed in xylene for 2 minutes. Neutral gum was added dropwise above the tissue after the transparent treatment, and the slide was slightly tilted. The cover was gently placed along the slope of the slide by hand to prevent bubbles.

## Statistics

The ggpubr package of R (version 3.5.2) was applied for statistical analysis and plotting. For Masson Staining, IPP6.0 software was used to measure the fibrotic area of Masson photos, and

three 400 × photos of each slice were selected for analysis (fibrosis area / tissue area = fibrosis proportion) to obtain the average fibrosis proportion representing the fibrosis level of the sample. For toluidine blue staining, three 400 × visual fields were selected for each sample. The number of all mast cells and the total number of cells in each visual field were counted, and the percentage of mast cells in each visual field was calculated. Data is presented as mean ± standard deviation. Wilcoxon test were used for comparisons between the two groups. $P < 0.05$ was considered as statistical significance.

## Results

### Case presentation

The clinical data of proband 1 was obtained to present on July 10, 2023 by applying to the Ethics Committee. Proband 1 was a 62-year-old woman suffering from chest distress, wheezing discomfort, obvious chest pain, suffocation discomfort, obvious dizziness, amaurosis, and decreased appetite for more than two years. She was then admitted to Wuhan Asia Heart Hospital and diagnosed with DCM, excluding common heart diseases such as coronary heart disease (CAD), ischemic cardiomyopathy, and rheumatic valvular heart disease. Echocardiography showed that the left heart was significantly enlarged, the motion amplitude of the interventricular septum and the left ventricular wall was generally significantly reduced, the mitral regurgitation was severe, and the left ventricular systolic function was significantly reduced (LV = 6.9 / 5.9 cm, EF = 24%) (Fig 1A and 1B). Also, the cardiac magnetic resonance imaging (MRI) displayed that the left heart was enlarged, the overall contraction of the interventricular septum and left ventricular wall was generally reduced. Additionally, myocardial delayed imaging exhibited myocardial fibrosis in the interventricular septum, left ventricular anterior wall and inferior wall. Myocardial perfusion imaging showed no significant ischemia in the myocardium, decreased left ventricular systolic function (EF value 22%), and normal right ventricular systolic function. Electrocardiogram was characterized as I II III aVL aVF V5-6 low flat or bidirectional, of which the diagnostic results were sinus rhythm, atrial premature beat and ST-T change (Fig 1C). After discussion by the ethics committee, the patient had indications for heart transplantation, which met the ethical requirements. Fortunately, heart transplantation and surgical cardiac pacemaker placement were successfully performed on the patient. After the operation, the patient was transferred to ICU, given ventilator assistance, maintenance of cardiac function, tube expansion, adjustment of blood volume, stabilization of internal environment, anti-infection and other treatments, and given triple immunosuppressant anti-rejection drug regimen. Whole exome sequencing (WES) revealed that proband 1 carried a versus uncertain significance (VUS) mutation *TTN* p. Tyr4418Ter verified by Sanger sequencing (Fig 1G') with no copy number variation (CNV) being found. According to the family history, her father died of cerebral hemorrhage, and her mother died of heart disease and repeated heart failure. Among the 6 siblings, the proband's sister also had heart disease. Since blood samples of family members were not available, pedigree verification could not be completed.

The clinical data of proband 2 was obtained to present on July 10, 2023 by applying to the Ethics Committee. Proband 2 was a 51-year-old woman once diagnosed with DCM at Wuhan Asia Heart Hospital in 2018. In March 2021, due to intermittent chest distress, precordium compression, and paroxysmal dyspnea at night for almost a month, proband 2 visited Wuhan Asia Heart Hospital seeking for treatment again and was hospitalized. As with the first diagnosis, based on the following examination results, the patient was diagnosed with dilated cardiomyopathy and myocardial infarction, and CAD was excluded. Echocardiography showed left heart enlargement, general reduction of the interventricular septum and left ventricular wall

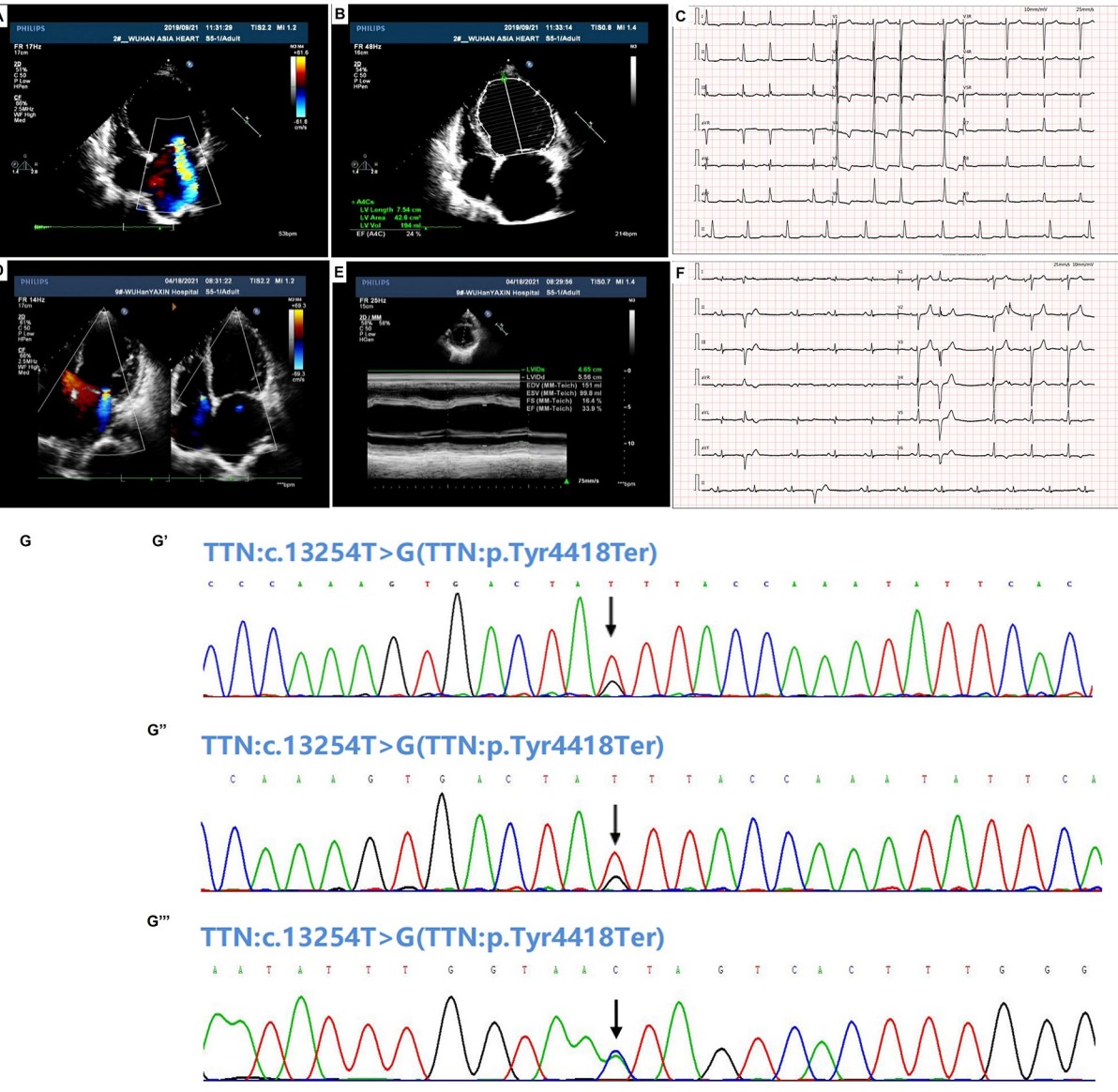

**Fig 1. Echocardiography, electrocardiogram and sanger sequencing results of the probands.** (A) and (B) Echocardiography of proband 1. (C) Electrocardiogram of proband 1. HR: 64bpm, P: 110ms, PR: 149ms, QRS: 104ms, QT/QTc: 440/450ms, P/QRS/T: 69/53/143deg, RV5/SV1: 2.176/1.602mV. (D) and (E) Echocardiography of proband 2. (F) Electrocardiogram of proband 2. HR: 64bpm, P: 125ms, PR: 187ms, QRS: 100ms, QT/QTc: 427/436ms, P/QRS/T: 57/75/129deg, RV5/SV1: 0.687/0.635mV. (G) Sanger sequencing results of TTN p. Tyr4418Ter mutation for proband 1 (G') and proband 2 (G"). (G"') The reverse sequencing result suggested that proband 2's daughter carries the mutation TTN p. Tyr4418Ter.

motion, moderate to severe mitral regurgitation, moderate to severe tricuspid valve regurgitation, pulmonary hypertension (moderate), and reduced left ventricular systolic function (LV = 5.6 / 4.4 cm, EF = 35%) (Fig 1D and 1E). Identically, cardiac MRI showed left ventricular enlargement, and segmental motion abnormality of left ventricular wall. Besides, myocardial perfusion imaging showed myocardial ischemia from the basal segment to the middle segment of the anterior wall of the left ventricle, the lower septum, and from the basal segment of the inferior wall to the apical segment of the endocardium. Furthermore, myocardial delayed imaging showed linear fibrosis in the middle layer of the interventricular septum from the

base to the middle segment, local myocardial fibrosis and microvascular occlusion in the lower septum and the lower wall apical segment, severe reduction of left ventricular systolic function, and moderate reduction of right ventricular systolic function. Electrocardiogram was characterized by I II III aVL aVF V5-6 low flat or bidirectional, of which the diagnostic results were sinus rhythm, left atrial abnormity, left ventricular hypertrophy, ventricular premature beat and T wave changes (Fig 1F). The proband was treated with anticoagulation, improving myocardial prognosis, strengthening heart, diuresis, nourishing myocardium and maintaining electrolyte balance. Whole exome sequencing showed that the proband carried the same VUS mutation *TTN* p. Tyr4418Ter that was verified by Sanger sequencing (Fig 1G"). No CNV was detected. The father of the proband 2 passed away at age 40 with unknown reason. Family verification indicated that her daughter also carries the pathogenic mutation (Fig 1G'''). The family tree couldn't be obtained due to the lack of samples of her family members.

## Mouse model

Off-target analysis results for gRNA-A1 are shown in Fig 2A. Through CRISPR/Cas-mediated genome engineering, 3 females (number #4, #6 and 9) and 1 male (number #2) heterozygous *TTN* KO mice were finally generated. All the 4 animals were further examined carrying *TTN* p.Y4370* mutant by Sanger sequencing (Fig 2B). In addition, 4 WT even-aged mice on a genetic background of the C57BL/6 strain, 3 females and 1 male, named L161, L162, L164, and L165 were chosen as the control.

## Serological detection results

Serological detection was executed 8 times in total. As shown in Fig 3A'–3A''', AST, LDH and CK of $TTN^{+/-}$ mice were significantly raised than those of WT mice in the second detection, indicating that myocardial injury presented.

## Echocardiography results

Echocardiography was performed a total of 5 times. The first echocardiograms of mice #2 and L161 were taken as examples shown in Fig 3B–3E. Among the indexes of the left ventricle obtained by the first echocardiography, EF and FS of the $TTN^{+/-}$ mice showed a downward trend compared with the WT mice (Table 1). As exhibited in Table 1, for the $TTN^{+/-}$ mice, the LV Mass significantly increased, the diastolic left ventricular anterior wall (LVAW) markedly thickened with a trend of thickening of systolic LVAW. Besides, the systolic left ventricular posterior wall (LVPW) and the diastolic LVPW of $TTN^{+/-}$ showed a trend of thickening. Regarding the pulmonary artery outflow bloodstream, the Peak Vel of $TTN^{+/-}$ mice in the third examination were apparently increased compared with WT mice (Table 2).

## Histological evaluation

Masson staining results of $TTN^{+/-}$ mouse #2 and WT mouse L162 were taken as examples. As shown in Fig 4A' and 4A", the collagen fibers were blue, the muscle fibers were red, and the nucleus was blue-black after Masson staining. Fig 4A''' illustrates that the cardiac fibrosis area of $TTN^{+/-}$ mice was significantly larger than that of WT mice. For toluidine blue staining, slices of $TTN^{+/-}$ mouse #6 (Fig 4B') and WT mouse L162 (Fig 4B") were taken as examples, exhibiting that the nucleus was blue, and the mast cells were metachromatic purplish red when encountering toluidine blue due to the presence of heterochromatic substances such as heparin and histamine in the cytoplasm. Wilcoxon test suggested that the positive rate of the cardiac mast cell of $TTN^{+/-}$ mice was significantly higher than that of WT mice (Fig 4B''').

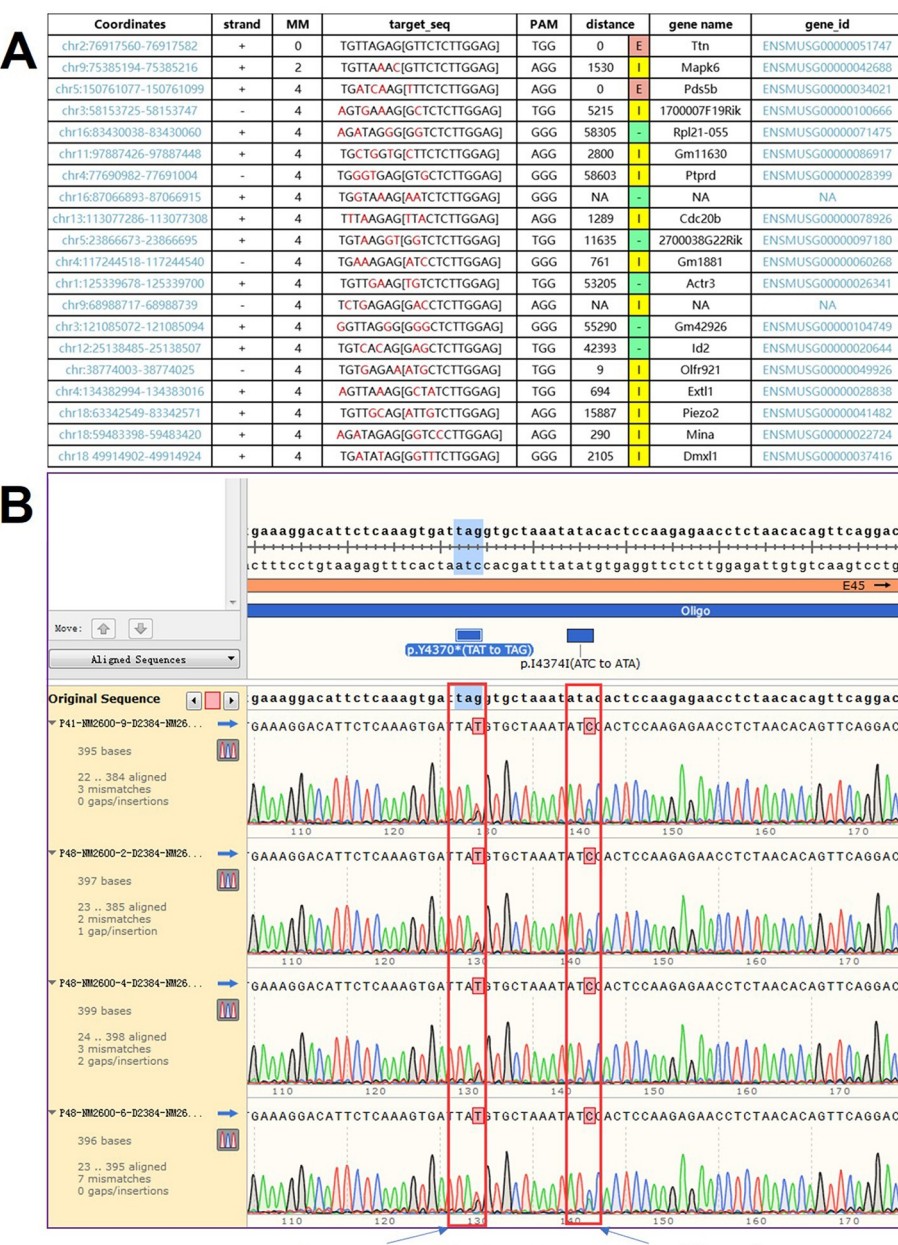

**Fig 2. Validation of the generated mouse model.** (A) Off-target analysis results for gRNA-A1. (B) Sanger sequencing validation results for the 4 TTN+/- mice. Each mouse carried the heterozygous nonsense mutation TTN p.Y4370*.

Besides, the heart weight of $TTN^{+/-}$ mice had an increasing trend compared with WT mice (Fig 4C' and 4C'').

## Discussion

Previously, we found two patients with dilated cardiomyopathy carrying a heterozygous *TTN* truncating mutant c.13254T>G which was recognized as a rare variant in population

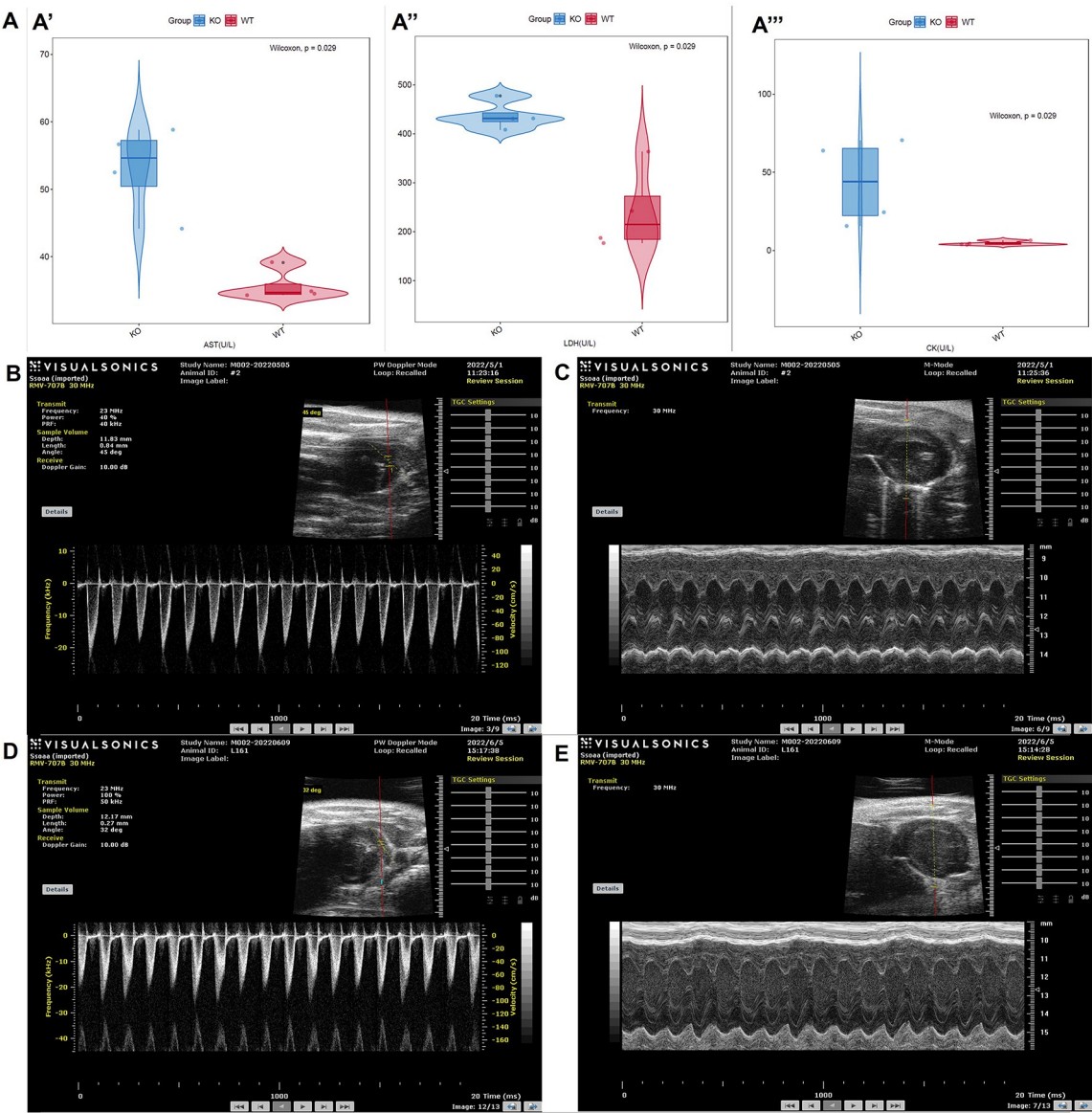

**Fig 3. Partial Results of serological detection and echocardiography.** (A) Results of the second serological detection. (B) Echocardiography of *TTN*^+/- mouse #2 pulmonary artery outflow tract. (C) Echocardiography of *TTN*^+/- mouse #2 left ventricle. (D) Echocardiography of L161 WT mouse pulmonary artery outflow tract. (E) Echocardiography of L161 WT mice left ventricle.

frequency databases (Thousand Human Genome:0.0001997; ESP6500: None; ExAC: $4.997e^{-05}$) and could not be searched out in ClinVar and HGMD databases. *TTN* c.13254T>G leads to an earlier termination of translation at amino acids 4418, which highly likely results in the expression of the truncating protein, and cardiomyopathy. Additionally, the variations c.10120A>T (p. Lys3374Ter), c.5383A>T (p. Lys1795Ter), and c.5047C>T (p. Arg1683Ter) downstream of c.13254T>G are already verified as causes of DCM. Thus, we assumed that *TTN* c.13254T>G was a pathogenic cause of DCM, and developed a mouse model of dilated cardiomyopathy carrying the mutant to supplement and improve the functional evidence of the rare variant *TTN* c.13254T>G.

**Table 1. Left ventricular function data statistics at 2-month time point.**

| Group | WT | | | | TTN[+/-] | | | | WT | TTN[+/-] | P |
|---|---|---|---|---|---|---|---|---|---|---|---|
| Study Name | L161 | L162 | L164 | L165 | #2 | #4 | #6 | #9 | Mean ± SD | | |
| Age (weeks) | 8 | 8 | 8 | 8 | 8 | 8 | 8 | 7 | 8±0 | 7.75±0.5 | 0.45 |
| Weight(g) | 17.9 | 16.8 | 16.5 | 25.3 | 24.7 | 17.9 | 18 | 19 | 19.13±4.16 | 19.9±3.24 | 0.38 |
| HR(bpm) | 500 | 501 | 486 | 490 | 513 | 501 | 541 | 495 | 494.25±7.41 | 512.5±20.42 | 0.15 |
| LVID;s(mm) | 1.9 | 1.4 | 1.3 | 1.2 | 1.57 | 1.02 | 2.19 | 1.6 | 1.45±0.32 | 1.6±0.48 | 0.69 |
| LVID;d(mm) | 3.4 | 2.9 | 3.1 | 3.1 | 2.87 | 2.61 | 3.68 | 3.06 | 3.12±0.2 | 3.06±0.46 | 0.49 |
| LV Vol;s(µL) | 8.01 | 3.97 | 4.08 | 3.49 | 5.64 | 2.32 | 13.51 | 5.65 | 4.89±2.1 | 6.78±4.75 | 0.69 |
| LV Vol;d(µL) | 51.33 | 36.88 | 39.66 | 40.95 | 32.69 | 22.95 | 60.54 | 36.29 | 42.21±6.32 | 38.12±15.98 | 0.34 |
| SV(µL) | 43.32 | 32.91 | 35.58 | 37.45 | 27.05 | 20.63 | 47.03 | 30.64 | 37.32±4.42 | 31.34±11.25 | 0.34 |
| EF(%) | 84.39 | 89.24 | 89.72 | 91.47 | 82.75 | 89.89 | 77.68 | 84.43 | 88.71±3.03 | 83.69±5.03 | 0.34 |
| FS(%) | 52.49 | 58.48 | 59.23 | 61.99 | 50.11 | 58.96 | 45.58 | 51.99 | 58.05±4 | 51.66±5.56 | 0.11 |
| CO(mL/min) | 21.66 | 14.45 | 17.29 | 18.35 | 13.88 | 10.34 | 24.69 | 15.17 | 17.94±2.98 | 16.02±6.13 | 0.49 |
| LV Mass(mg) | 92.68 | 103.46 | 89.29 | 127.72 | 160 | 131.27 | 181.91 | 157.95 | 103.29±17.37 | 157.78±20.74 | 0.029 |
| LV Mass (Corrected,mg) | 74.14 | 82.76 | 71.43 | 102.17 | 128 | 105.02 | 145.53 | 126.36 | 82.63±13.9 | 126.23±16.59 | 0.029 |
| LVAW;s(mm) | 1.44 | 2.04 | 1.66 | 2.04 | 2.08 | 2.06 | 1.95 | 2.15 | 1.8±0.3 | 2.06±0.08 | 0.11 |
| LVAW;d(mm) | 0.86 | 1.29 | 1.08 | 1.27 | 1.52 | 1.35 | 1.49 | 1.65 | 1.13±0.2 | 1.5±0.12 | 0.029 |
| LVPW;d(mm) | 0.78 | 0.77 | 0.67 | 0.98 | 1.21 | 1.27 | 0.88 | 0.92 | 0.8±0.13 | 1.07±0.2 | 0.11 |
| LVPW;s(mm) | 1.17 | 1.21 | 1.29 | 1.58 | 1.74 | 1.8 | 1.7 | 1.36 | 1.31±0.19 | 1.65±0.2 | 0.057 |

HR: Heart Rate; LVID;S: Left Ventricular Systolic Internal Diameter; LVID;D: Left Ventricular Diastolic Internal Diameter; LV Vol;S: Left Ventricular End Systolic Volume; Left Ventricular Vol;D: Left Ventricular End Diastolic Volume; SV: Left Ventricular Stroke Volume; EF: Ejection Fraction; FS: Fractional Shortening; CO: Cardiac Output; LV Mass: Left Ventricular Mass; LVAW;S: Left Ventricular Systolic Anterior Wall; LVAW;D: Left Ventricular Diastolic Anterior Wall; LVPW;D: Left Ventricular Diastolic Posterior Wall; LVPW;S: Left Ventricular Systolic Posterior Wall.

At 2 months old, the myocardial enzyme indicators AST, LDH and CK of the *TTN*[+/-] mice were significantly higher than those of the WT mice, indicating that myocardial injury may occurred in the *TTN*[+/-] mice. In consistency with the consequence that the *TTN*[+/-] mice presented myocardial injury, the first echocardiography displayed a thickened diastolic LVAW, an aggravated LV mass in the *TTN*[+/-] mice with a trend of reduced EF and FS, suggesting that cardiac structure altered and cardiac function was affected. The third echocardiography displayed an obvious increase in Peak Vel of *TTN*[+/-] mice, showing that pulmonary valve stenosis and pulmonary aortic stenosis may appear in 6-month-old *TTN*[+/-] mice. In addition, cardiac fibrosis level and positive rate of cardiac mast cell of TTN+/- mice were obviously increased compared with WT mice, and the heart weight of *TTN*[+/-] mice had an increasing trend, manifesting the abnormal trend of *TTN*[+/-] mice's myocardial structure. We hypothesize that the

**Table 2. Pulmonary artery outflow bloodstream data statistics at 6-month time point.**

| Group | WT | | | | TTN[+/-] | | | | WT | TTN[+/-] | P |
|---|---|---|---|---|---|---|---|---|---|---|---|
| Study Name | L161 | L162 | L164 | L165 | #2 | #4 | #6 | #9 | Mean ± SD | | |
| PV VTI(cm) | 4.39 | 4.83 | 4.79 | 5.1 | 4.75 | 5.12 | 5.33 | 5.49 | 4.78±0.29 | 5.17±0.32 | 0.2 |
| Mean Vel(mm/s) | 751 | 725.98 | 683.25 | 833.14 | 848.98 | 871.73 | 797.97 | 792.51 | 748.34±63.07 | 827.8±38.79 | 0.11 |
| Mean Grad(mmHg) | 2.27 | 2.11 | 1.87 | 2.82 | 2.9 | 3.04 | 2.55 | 2.52 | 2.27±0.4 | 2.75±0.26 | 0.11 |
| Peak Vel(mm/s) | 1146.42 | 1197.68 | 1022.66 | 1269.82 | 1299.41 | 1377.09 | 1290.16 | 1269.82 | 1159.15±104.12 | 1309.12±46.97 | 0.042 |
| Peak Grad(mmHg) | 5.28 | 5.78 | 4.18 | 6.48 | 6.76 | 7.62 | 6.68 | 6.47 | 5.43±0.97 | 6.88±0.51 | 0.057 |
| PAT(ms) | 11.25 | 11.88 | 14.38 | 15.63 | 11.25 | 12.5 | 12.5 | 13.75 | 13.29±2.07 | 12.5±1.02 | 0.77 |

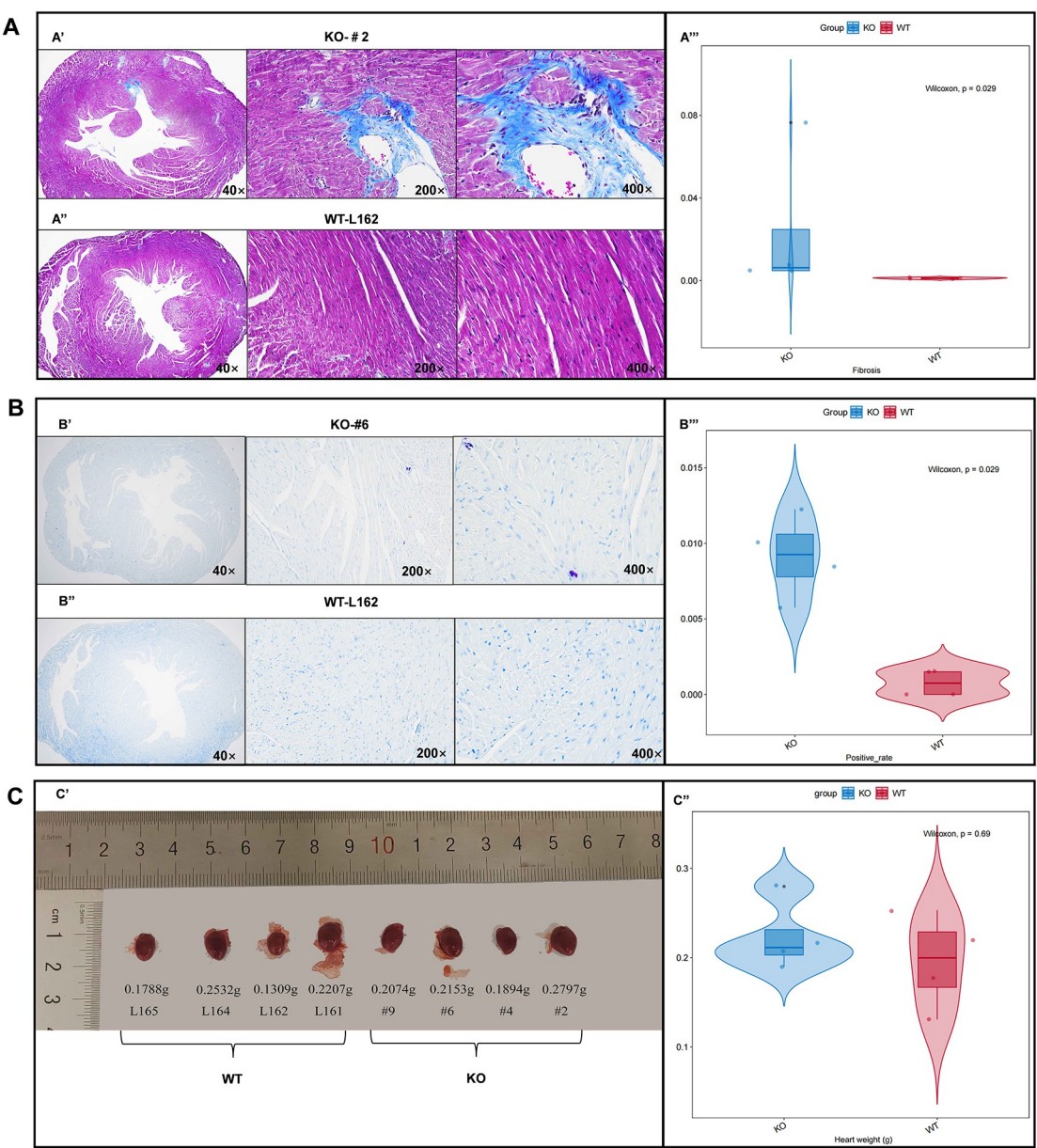

**Fig 4. Histological evaluation for mice heart.** (A') Masson staining results of $TTN^{+/-}$ mouse #2 heart tissue section under 40×, 200× and 400× magnification. (A") Masson staining results of WT mouse L161 heart tissue section under 40×, 200× and 400× magnification. (A"') Wilcoxon test indicated that cardiac fibrosis area of $TTN^{+/-}$ mice was significantly larger than that of WT mice. (B') Toluidine blue staining results of $TTN^{+/-}$ mouse #6 heart tissue section under 40×, 200× and 400× magnification. (B") Toluidine blue staining results of WT mouse L162 heart tissue section under 40×, 200× and 400× magnification. (B"') Wilcoxon test suggested that the cardiac mast cell positive rate of $TTN^{+/-}$ mice was significantly higher than that of WT mice. (C') Images showing heart size and mass of WT and $TTN^{+/-}$ mic. (C") $TTN^{+/-}$ mice showed an increasing trend in the heart weight compared with WT mice.

*TTN* p.Y4370* mutation causes alterations in the sarcomere structure, leading to abnormal results of those examinations. The giant protein Titin encoded by *TTN* is an important component of the sarcomere, the largest protein in human body and the third largest striated muscle protein, which is composed of about 33000 amino acids. The molecular structure of human Titin consists of four different parts, the Z disk, I band, A band and the M line at the carboxyl

end. Two main full-length Titin subtypes N2BA and N2B are stably expressed in adult myocardium, along with the low content short Novex subtype [11]. N2BA and N2B typically exist in a ratio of 70:30–60:40 [17]. Compared to the shorter and harder N2B subtype, the N2BA subtype is slightly longer and more flexible. Titin plays an important role in assembling the sarcomere [18], sensing mechanical stimulation and converting it into biochemical signals, providing passive tension in striated muscle, and regulating the active contractility of the sarcomere [19]. N2B and N2BA are co-expressed in sarcomeres and both participate in the mechanical sensing and signal transduction pathways of myocardial fibers [20]. As long as the ratio of these two subtypes changes slightly, there will be a significant change in the passive tension of the myocardium, and there will also be a significant difference in the stiffness of the myocardium. In the bovine model experiment of heart failure caused by rapid pacing, the total amount of Titin in the left ventricular myocardium did not decrease, but the N2B subtype relatively increased, accompanied by an increase in passive stiffness [20]. However, the pathogenicity of TTNtv c.13254T>G is on transcript mRNA NM_133379.5, which translates into the protein novex-3. The transcript mRNA of N2B and N2BA are NM_003319.4 and NM_001256850.01, respectively. Therefore, whether TTNtv c.13254T>G leads to co-expression of TTN protein subtypes needs to be further studied. Nagueh et al. [17] reported that the N2BA/N2B ratio in DCM patients was higher than that in the control group, suggesting that the change of N2BA/N2B subtype ratio caused by Titin gene mutation may be the cause of DCM.

There are several mechanisms have been proposed to elucidate the pathogenic process of *TTN*tv. First, the "toxic peptide" effect. *TTN*tv causes abnormal shortening of *TTN* proteins, leading to the loss of some important functional structures and the formation of "toxic peptides", ultimately leading to related dysfunction [9]. For example, the A-band is a β-junction point between myosin and coarse myofilaments, and its interaction with muscle ring finger protein 2 (MuRF2) promotes the formation and maturation of sarcomere. Studies have shown that *TTN*tv in the A-band region leads to the loss of β-myosin binding sites, which affects sarcomere assembly and myocardial systolic function [21]. In addition, the M line of *TTN* is related to the perception and regulation of sarcomere force, and *TTN*tv occurring at the carboxyl end causes the M line of *TTN* to be truncated, causing muscle strength regulation disorders, leading to implicit and early-onset DCM. Second, insufficient single dose and cardiac metabolic disorders. Contrary to the dominant negative effect, *TTN*tv in the A-band and I-band regions can avoid abnormal *TTN* production through nonsense-mediated mRNA degradation (NMD). Subsequent studies found that a decrease in the number of *TTN*s in cells can reduce the heart's metabolism of medium and long-chain fatty acids, and increase the heart's dependence on glycolysis. The molecular mechanism is still unclear [22]. The long-term elevation of glycolytic intermediates and branched-chain amino acids can lead to the activation of the serine/threonine protein kinase mammalian rapamycin target protein complex 1 (mTORC1) signaling pathway, thereby promoting ineffective protein synthesis and autophagy, leading to myocardial cell damage and cardiac contraction disorders [23]. In addition, Adams et al. [24] found that *TTN*tv can affect the electron transfer of the heart mitochondrial respiratory chain, and then lead to the increase of the level of reactive oxygen species (ROS) and the increase of mitochondrial protein ubiquitination. These factors can lead to myocardial structural damage and insufficient energy supply through autophagy and mitochondrial oxidative phosphorylation disorder, and eventually lead to myocardial contraction disorder and the formation of DCM. In this study, the transcript mRNA NM_003319.4 has 5604 amino acids, of which TTNtvc.13254T>G resulted in an advance stop codon at the 4418 amino acid position, resulting in protein truncation. In addition, the downstream loci c.10120A>T (p.Lys3374Ter), c.5383A>T (p.Lys1795Ter), and c.5047C>T (P.RG1683ter) of *TTN* gene have all been reported as pathogenic mutations of dilated cardiomyopathy in Clinvar (NM_001267550.2),

so the TTNtv c.13254T>G may lead to TTN protein truncation changes, and then lead to dilated cardiomyopathy.

In this study, we reported two patients who were clinically diagnosed with DCM carrying the same *TTN*tv c.13254T>G discovered by WES technology and generated a mouse model with mutation *TTN* p.Y4370*. Results of serological detection, echocardiography, and histological evaluation demonstrated that the structure and function of cardiac abnormalities occurred in *TTN*$^{+/-}$ mice, supplementing and explaining the pathogenicity evidence of *TTN*tv c.13254T>G.

## Author Contributions

**Conceptualization:** Hongyan Xiao.

**Data curation:** Wenqiang Sun.

**Formal analysis:** Wenqiang Sun, Xiaohui Liu, Kaisheng Lai.

**Funding acquisition:** Wenqiang Sun.

**Investigation:** Xiaohui Liu, Liang Tao, Kaisheng Lai, Hui Jiang.

**Project administration:** Hongyan Xiao.

**Resources:** Laichun Song, Kaisheng Lai.

**Software:** Wenqiang Sun, Liang Tao, Hui Jiang.

**Supervision:** Liang Tao, Hongyan Xiao.

**Writing – original draft:** Wenqiang Sun.

**Writing – review & editing:** Laichun Song, Kaisheng Lai, Hui Jiang, Hongyan Xiao.

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
