## [Decision Letter · Decision Letter 0]

29 Nov 2023

PONE-D-23-30447The TTN p. Tyr4418Ter mutation causes cardiomyopathy in human and micePLOS ONE

Dear Dr. Xiao,

Thank you for submitting your manuscript to PLOS ONE. After careful consideration, we feel that it has merit but does not fully meet PLOS ONE’s publication criteria as it currently stands. Therefore, we invite you to submit a revised version of the manuscript that addresses the points raised during the review process.

We look forward to receiving your revised manuscript.

Kind regards,

Nejat Mahdieh

Academic Editor

PLOS ONE

Journal Requirements:

This study is supported by Hubei Provincial Natural Science Foundation of China(2022CFB841).

4. Please include your tables as part of your main manuscript and remove the individual files. Please note that supplementary tables (should remain/ be uploaded) as separate "supporting information" files.

Reviewers' comments:

Reviewer's Responses to Questions

**Comments to the Author**

1. Is the manuscript technically sound, and do the data support the conclusions?

Reviewer #1: Yes

Reviewer #2: Yes

Reviewer #3: Yes

2. Has the statistical analysis been performed appropriately and rigorously? 

Reviewer #1: N/A

Reviewer #2: Yes

Reviewer #3: Yes

3. Have the authors made all data underlying the findings in their manuscript fully available?

Reviewer #1: Yes

Reviewer #2: Yes

Reviewer #3: Yes

4. Is the manuscript presented in an intelligible fashion and written in standard English?

Reviewer #1: Yes

Reviewer #2: Yes

Reviewer #3: Yes

5. Review Comments to the Author

Reviewer #1: The manuscript entitled "The TTN p. Tyr4418Ter mutation causes cardiomyopathy in human and mice" explains the A nonsense TTNtv c.13254T>G (TTN p. Tyr4418Ter) which hasn’t been recorded in general populations. they constructed a C57BL/6J mouse model carrying Y4370* mutation by CRISPR/Cas-mediated genome engineering to imitate the newly discovered human TTNtv followed by clinical examinations, including echocardiography, serological detection and histological evaluation. At 2 months old, the myocardial enzyme indicators AST, LDH and CK of the TTN+/- mice were significantly higher than those of the WT mice, indicating that myocardial injury may occurred in the TTN+/- mice. They also concluded that cardiac structure altered and cardiac function was affected. Thanks to the authors.

Comments to review

Did the authors analyzed the N2B and N2BA co-expression in sarcomeres?

Do the authors suggest and point to "toxic peptide" effect that causes important functional structures? Please clarify

Where does this variant affect protein structure /where is it located? Please explain the major cause of damage. Or the authors want to say all the factors are possible? Please clarify.

Reviewer #2: I will recommend rejection of this submission due to the fact that PLoS One criteria for publication is that "Results reported have not been published elsewhere" and "All figures included in manuscripts should be original, and should not have been published in any previous publications." and "we will not consider submissions that are currently under consideration for publication elsewhere."

As can be seen in https://papers.ssrn.com/sol3/papers.cfm?abstract_id=4527820, a report of the research reported in the paper submitted to PLoS ONE has already been submitted for publication elsewhere, with the same figures.

Reviewer #3: The methodology employed in this study is robust and well-designed. The clarity of the author's argumentation and the logical flow of ideas make the manuscript a pleasure to read. I appreciate the innovative approach the author has taken in addressing how TTNtv Y4370* could lead to alterations in cardiac structure and function in mice, supplementing the evidence of TTNtv c.13254T>G pathogenicity in humans.

6. PLOS authors have the option to publish the peer review history of their article (what does this mean?). If published, this will include your full peer review and any attached files.

Reviewer #1: No

Reviewer #2: **Yes: **Jose Carlos Pinto Barreto Ferreira

Reviewer #3: **Yes: **S.Justin Carlus

---

## [Author Response · Author response to Decision Letter 0]

3 Dec 2023

Dear editors and reviewers:

Thank you for your letter and for the reviewers’ comments concerning our manuscript entitle “The TTN p. Tyr4418Ter mutation causes cardiomyopathy in human and mice” (PONE-D-23-30447). Those comments are all valuable and very helpful for revising and improving our paper, as well as the important guiding significance to our researches. We online submitted a rebuttal letter, a marked-up copy of the manuscript that highlights changes, an unmarked version of the revised paper, a list of responses (with one point by point response), a new cover letter and four figures processed through PACE. We have studied comments carefully and have made correction which we hope meet with approval. Revised portion are highlighted in yellow in the paper. The main corrections in the paper and the responds to the reviewer’s comments are as following: 

Responds and rebuttal to the comments of academic editor and reviewer(s):

Journal Requirements:

1. When submitting your revision, we need you to address these additional requirements. Please ensure that your manuscript meets PLOS ONE's style requirements, including those for file naming. 

Response: Thanks for your important comments. We have examined our manuscript carefully, and the manuscript meets PLOS ONE's style requirements, including those for file naming.

2. Thank you for stating in your Funding Statement: This study is supported by Hubei Provincial Natural Science Foundation of China(2022CFB841). Please provide an amended statement that declares all the funding or sources of support (whether external or internal to your organization) received during this study, as detailed online in our guide for authors at http://journals.plos.org/plosone/s/submit-now. Please also include the statement “There was no additional external funding received for this study.” in your updated Funding Statement. Please include your amended Funding Statement within your cover letter. We will change the online submission form on your behalf.

Response: Thanks for your important comments. We have provided an amended statement that declares all the funding or sources of support (whether external or internal to your organization) received during this study. The statement “There was no additional external funding received for this study.” Is in the updated Funding Statement. We have included an amended Funding Statement within the cover letter.

3. PLOS requires an ORCID iD for the corresponding author in Editorial Manager on papers submitted after December 6th, 2016. Please ensure that you have an ORCID iD and that it is validated in Editorial Manager. To do this, go to ‘Update my Information’ (in the upper left-hand corner of the main menu), and click on the Fetch/Validate link next to the ORCID field. This will take you to the ORCID site and allow you to create a new iD or authenticate a pre-existing iD in Editorial Manager. Please see the following video for instructions on linking an ORCID iD to your Editorial Manager account: https://www.youtube.com/watch?v=_xcclfuvtxQ.

Thanks for your important comments. 

Response: We have provided an ORCID iD of the corresponding author.

4. Please include your tables as part of your main manuscript and remove the individual files. Please note that supplementary tables (should remain/ be uploaded) as separate "supporting information" files.

Response: Thanks for your important comments. We have included tables as part of your main manuscript and remove the individual files. The supplementary tables have been uploaded as separate "supporting information" files.

Comments to the Author:

1. Is the manuscript technically sound, and do the data support the conclusions? The manuscript must describe a technically sound piece of scientific research with data that supports the conclusions. Experiments must have been conducted rigorously, with appropriate controls, replication, and sample sizes. The conclusions must be drawn appropriately based on the data presented. 

Reviewer #1: Yes

Reviewer #2: Yes

Reviewer #3: Yes

Response: Thanks for your important comments. Thank you very much.

2. Has the statistical analysis been performed appropriately and rigorously?

Reviewer #1: N/A

Reviewer #2: Yes

Reviewer #3: Yes

Response: Thanks for your important comments. We have examined our statistical analysis program of the manuscript carefully, and make sure that the statistical analysis appropriately and rigorously.

3. Have the authors made all data underlying the findings in their manuscript fully available? The PLOS Data policy requires authors to make all data underlying the findings described in their manuscript fully available without restriction, with rare exception (please refer to the Data Availability Statement in the manuscript PDF file). The data should be provided as part of the manuscript or its supporting information, or deposited to a public repository. For example, in addition to summary statistics, the data points behind means, medians and variance measures should be available. If there are restrictions on publicly sharing data—e.g. participant privacy or use of data from a third party—those must be specified.

Reviewer #1: Yes

Reviewer #2: Yes

Reviewer #3: Yes

Response: Thanks for your important comments. Thank you very much.

4. Is the manuscript presented in an intelligible fashion and written in standard English? PLOS ONE does not copyedit accepted manuscripts, so the language in submitted articles must be clear, correct, and unambiguous. Any typographical or grammatical errors should be corrected at revision, so please note any specific errors here.

Reviewer #1: Yes

Reviewer #2: Yes

Reviewer #3: Yes

Response: Thanks for your important comments. Thank you very much.

5. Review Comments to the Author:

Reviewer #1

Reviewer #1: The manuscript entitled "The TTN p. Tyr4418Ter mutation causes cardiomyopathy in human and mice" explains the A nonsense TTNtv c.13254T>G (TTN p. Tyr4418Ter) which hasn’t been recorded in general populations. they constructed a C57BL/6J mouse model carrying Y4370* mutation by CRISPR/Cas-mediated genome engineering to imitate the newly discovered human TTNtv followed by clinical examinations, including echocardiography, serological detection and histological evaluation. At 2 months old, the myocardial enzyme indicators AST, LDH and CK of the TTN+/- mice were significantly higher than those of the WT mice, indicating that myocardial injury may occurred in the TTN+/- mice. They also concluded that cardiac structure altered and cardiac function was affected. Thanks to the authors.

Response: Thanks for your important comments. Thank you very much.

Comments to review：

Did the authors analyzed the N2B and N2BA co-expression in sarcomeres? Do the authors suggest and point to "toxic peptide" effect that causes important functional structures? Please clarify Where does this variant affect protein structure /where is it located? Please explain the major cause of damage. Or the authors want to say all the factors are possible? Please clarify.

Response: Thanks for your important comments. First, we did not do co-expression analysis of N2B and N2BA in this study, because the pathogenicity of TTNtv c.13254T>G is on transcript mRNA NM_133379.5. However, the transcript mRNA of N2B and N2BA are NM_003319.4 and NM_001256850.01, respectively. Besides, the TTNtv c.13254T>G locates in the intron region on transcript mRNA NM_003319.4 and NM_001256850.01. Although we did not analyze the co-expression of N2B and N2BA, we are conducting transcriptional expression analysis of TTN gene in the heart tissue of mouse models to further confirm the protein truncation changes caused by TTNtvc.13254T>G，and whether TTNtv c.13254T>G leads to co-expression of TTN protein subtypes needs to be further studied. (Page 19, line 20-23. Page 20, line1)

Second, we really agree with the opinion of the reviewer, point to toxic peptide effect that causes important functional structures. The transcript mRNA NM_003319.4 has 5604 amino acids, of which TTNtvc.13254T>G resulted in an advance stop codon at the 4418 amino acid position, resulting in protein truncation. In addition, the downstream loci c.10120A>T (p.Lys3374Ter), c.5383A>T (p.Lys1795Ter), and c.5047C>T (P.RG1683ter) of TTN gene have all been reported as pathogenic mutations of dilated cardiomyopathy (NM_001267550.2), so the TTNtv c.13254T>G can lead to TTN protein truncation changes, and then lead to dilated cardiomyopathy. (Page 21, line 7-13) We have marked and cited relevant research literature evidence in the manuscript. (Page 19, line 13-14)

Reviewer #2

Reviewer #2: I will recommend rejection of this submission due to the fact that PLoS One criteria for publication is that "Results reported have not been published elsewhere" and "All figures included in manuscripts should be original, and should not have been published in any previous publications." and "we will not consider submissions that are currently under consideration for publication elsewhere." As can be seen in https://papers.ssrn.com/sol3/papers.cfm?abstract_id=4527820, a report of the research reported in the paper submitted to PLoS ONE has already been submitted for publication elsewhere, with the same figures.

Response: Thanks for your important comments. We are sorry for this manuscript could be seen in https://papers.ssrn.com/sol3/papers.cfm?abstract_id=4527820. We have always believed that this manuscript was not published elsewhere. We have carefully reviewed the submission process and believe we made a mistake that led to the manuscript preprint publication at the same time we submitted the submission. We will attempt to withdraw the manuscript from the preprint publishing system. We are sorry for this mistake again.

Reviewer #3

Reviewer #3: The methodology employed in this study is robust and well-designed. The clarity of the author's argumentation and the logical flow of ideas make the manuscript a pleasure to read. I appreciate the innovative approach the author has taken in addressing how TTNtv Y4370* could lead to alterations in cardiac structure and function in mice, supplementing the evidence of TTNtv c.13254T>G pathogenicity in humans.

Response: Thanks for your important comments. Thank you very much.

Special thanks to you for your good comments.

We tried our best to improve the manuscript and made some changes in the manuscript. According to the comments of reviewers, we have revised the manuscript and list specific changes here point by point. The changes have been highlighted in yellow in revised paper. The table line number of specific changes is based on the new revised manuscript.

We appreciate for Editors/Reviewers’ warm work earnestly, and hope that the correction will meet with approval. 

Once again, thank you very much for your comments and suggestions. 

The list of specific changes

1. Page 11, line 22-23:

 The sentence “Echocardiography, electrocardiogram and sanger sequencing results of the probands” was added.

2. Page 12, line 15:

 The sentence “Validation of the generated mouse model.” was added.

3. Page 12, line 22:

 The sentence “Partial Results of serological detection and echocardiography.” was added.

4. Page 19, line 13-14:

The refference “Wu Y, Bell SP, Trombitas K, Witt CC, Labeit S, LeWinter MM, et al. Changes in titin isoform expression in pacing-induced cardiac failure give rise to increased passive muscle stiffness. Circulation. 2002;106(11):1384-9. Epub 2002/09/11. doi: 10.1161/01.cir.0000029804.61510.02. PubMed PMID: 12221057.” was added.

5. Page 19, line 20-23. Page 20, line1:

 The sentences “However, the pathogenicity of TTNtv c.13254T>G is on transcript mRNA NM_133379.5, which translates into the protein novex-3. The transcript mRNA of N2B and N2BA are NM_003319.4 and NM_001256850.01, respectively. Therefore, whether TTNtv c.13254T>G leads to co-expression of TTN protein subtypes needs to be further studied.” were added.

6. Page 21, line 7-13:

The sentences “In this study, the transcript mRNA NM_003319.4 has 5604 amino acids, of which TTNtvc.13254T>G resulted in an advance stop codon at the 4418 amino acid position, resulting in protein truncation. In addition, the downstream loci c.10120A>T (p.Lys3374Ter), c.5383A>T (p.Lys1795Ter), and c.5047C>T (P.RG1683ter) of TTN gene have all been reported as pathogenic mutations of dilated cardiomyopathy in Clinvar (NM_001267550.2), so the TTNtv c.13254T>G may lead to TTN protein truncation changes, and then lead to dilated cardiomyopathy.” were added.

7. Page 21, line 22:

The sentences “There was no additional external funding received for this study.” was added.

8. Page 22, line 18-21 and page 23, line 1-3:

The author contributions were corrected as “Hongyan Xiao: Conceptualization, Supervision, Funding acquisition, Writing - Review & Editing; Resources; Wenqiang Sun: Investigation, Formal analysis, Methodology, Writing - original draft, Writing - review & editing; Xiaohui Liu: Investigation, Formal analysis, Software, Writing - original draft, Writing - review & editing; Laichun Song: Data curation, Validation, Writing - review & editing; Liang Tao: Project administration, Visualization; Kaisheng Lai: Resources, Writing - review & editing; Hui Jiang: Resources, Writing - review & editing.”.

---

## [Editor Report · Decision Letter 1]

19 Dec 2023

The TTN p. Tyr4418Ter mutation causes cardiomyopathy in human and mice

PONE-D-23-30447R1

Dear Dr. Xiao,

We’re pleased to inform you that your manuscript has been judged scientifically suitable for publication and will be formally accepted for publication once it meets all outstanding technical requirements.

Kind regards,

Nejat Mahdieh

Academic Editor

PLOS ONE
---

## [Editor Report · Acceptance letter]

9 Feb 2024

PONE-D-23-30447R1 

PLOS ONE

Dear Dr. Xiao, 

I'm pleased to inform you that your manuscript has been deemed suitable for publication in PLOS ONE. Congratulations! Your manuscript is now being handed over to our production team.

Kind regards, 

on behalf of

Dr. Nejat Mahdieh 

Academic Editor

PLOS ONE